

# Intra-rater and inter-rater reliability of a handheld myotonometer measuring myofascial stiffness of lower lumbar myofascial tissue in healthy adults

Fabio Valenti[1], Sara Meden[2], Maja Frangež[3] and Renata Vauhnik[4,5]

[1] Biotechnical Faculty, University of Ljubljana, Ljubljana, Slovenia
[2] Valdoltra Orthopaedic Hospital, Ankaran, Slovenia
[3] Institute for Medical Rehabilitation, University Medical Centre, Ljubljana, Slovenia
[4] Faculty of Health Sciences, Department of Physiotherapy, University of Ljubljana, Ljubljana, Slovenia
[5] Institute for Joints and Sport Injuries, ARTHRON, Ljubljana, Slovenia

## ABSTRACT

**Background:** Biomechanical muscle properties, such as stiffness, can be valuable indicators of tissue health and show promise as a diagnostic and treatment measure for chronic low back pain (CLBP). The development of accessible assessment technologies, such as the MyotonPRO portable device, allows for the convenient quantification of muscle tone and stiffness changes. The aim of this study is to assess the reliability of lumbar erector spinae muscle stiffness with the MyotonPRO in healthy adults and to compare stiffness changes between prone and sitting position.

**Methods:** Thirty asymptomatic participants ($N = 15$ women and $N = 15$ men) aged between 18 and 65 years were recruited to participate in this study. Two examiners tested muscle stiffness at the palpable muscle belly, one finger away from the spinous process at the level of the L4 vertebra, first from the left and then from the right side, both in prone position and after in sitting position. For inter-rater reliability, all participants were tested by two examiners on the same day, and intra-rater reliability was calculated using the same examiner's assessment results with an exact 24-h interval. Intraclass correlation coefficients (ICC), standard error measures (SEM) and minimum detectable change (MDC) with a 95% confidence interval were calculated to assess intra-rater and inter-rater reliability.

**Results:** Statistical analysis revealed good intra-rater reliability with an ICC of 0.88 (95% CI [0.76–0.94]) for the stiffness of the left erector spinae and excellent intra-rater reliability with an ICC of 0.91 (95% CI [0.82–0.95]) for the right erector spinae, both in the prone position. Intra-rater reliability in the sitting position was excellent to very good with an ICC of 0.91 (95% CI [0.82–0.96]) for the left side and an ICC of 0.89 (95% CI [0.78–0.95]) for the right side. The results for the left-sided prone position showed good inter-rater reliability with an ICC of 0.87 (95% CI [0.73–0.94]). The prone position on the right side also showed good inter-rater reliability with an ICC of 0.84 (95% CI [0.68–0.92]). The inter-rater reliability for the left and right side in the sitting position was excellent with an ICC of 0.96 (95% CI [0.92–0.98]) for the left side and an ICC of 0.95 (95% CI [0.90–0.97]) for the right side.

Corresponding author
Renata Vauhnik,
renata.vauhnik@zf.uni-lj.si

**Conclusion:** This study demonstrated high reliability in measuring lumbar erector spinae muscle stiffness with the MyotonPRO in healthy adults and the ability of the device to detect even small changes in erector spinae muscle stiffness, testing both the right and left sides and measuring in both prone and sitting positions. The use of the sitting position to assess lumbar tissue tension in individuals may serve as a valuable substitute for the prone position, particularly for patients who experience discomfort in the prone position, and could have additional practical significance in clinical settings.

## INTRODUCTION

According to the World Health Organisation (WHO), musculoskeletal disorders are a major global health problem and one of the main causes of disability (*WHO, 2023*). According to a recent Global Burden of Disease study, around 1.71 billion people worldwide are affected by musculoskeletal disorders (*Cieza et al., 2020*). Non-communicable diseases account for almost a fifth of all cases in Slovenia and contribute to 15.6% of the total burden in terms of years lived with disability (*Institute for Health Metrics and Evaluation (IHME), 2022*).

Low back pain (LBP) is a common musculoskeletal disorder that affects up to 90% of people at some point, with more than 50% experiencing recurrent episodes (*Arya, 2014*). Slovenia has the highest prevalence of LBP in Europe at 40.7% (*European Musculoskeletal Network, 2013*), while LBP is the second most common cause of work absenteeism (*Sirbu et al., 2020*). The prevalence of LBP increases with age, sedentary lifestyle and social status, emphasising the need for targeted interventions and preventive measures.

Biomechanical muscle properties, such as stiffness, are objective parameters that indicate the condition of the tissue and could become a reference for the diagnosis and treatment of CLBP (*Lohr et al., 2018*). The modulation of the stiffness of the erector spinae muscle of the lumbar spine remains unclear. Further research on the biomechanical properties of the lumbar muscles is needed to identify the activation mechanism in CLBP. The biomechanical properties of the muscles have been quantified using new technologies such as magnetic resonance elastography (MRE) and shear wave elastography (SWE) (*Hong et al., 2016*; *Koppenhaver et al., 2019*). However, these technologies have inherent limitations due to complex methodological procedures, image readout duration, transportability, accessibility and cost-effectiveness (*Li et al., 2022*).

Quantifying changes in paraspinal muscle tone and stiffness in the clinical setting remains a challenge. An affordable and practical assessment technology such as the MyotonPRO® portable, non-invasive handheld digital device has been developed to measure lumbar muscle stiffness. *Lee, Kim & Lee (2021)* demonstrated moderate to good correlation between the SWE and the MyotonPRO device when measuring muscle stiffness in the lower limbs of healthy subjects. Measurements were taken both at rest and during an

active voluntary contraction. It should be noted that both the MRE and SWE are reliable measurement tools, but they are also expensive devices that require skilled operators, which makes these techniques less feasible for use in clinics and sports setting, as may be the case with the MyotonPRO. This device applies a short (15 ms) low intensity (0.58 N) mechanical impulse to the skin and records the oscillatory tissue response. The internal software then calculates the resting tension, elasticity and stiffness of the tissue based on an acceleration curve (*Bizzini & Mannion, 2003*). The reliability and validity of the MyotonPRO has been studied in different populations to measure the viscoelastic properties of different skeletal muscle groups in healthy adults and individuals with pathological conditions. This includes assessing the properties of the lumbar and back muscles and myofascial tissue (*Xu, Hug & Fu, 2018*; *Wu et al., 2022*; *Nair et al., 2016*). The MyotonPRO has good to excellent reliability in healthy skeletal muscles (*Chen et al., 2019*). It has also been shown to reliably measure changes in superficial lumbar myofascial stiffness to a depth of 2 cm in healthy individuals and those with lumbalgia (*Grześkowiak, Kocur & Łochyński, 2023*). The myotonometer has also shown acceptable reliability when used in a clinical setting in young and older adults with CLBP (*Xu, Hug & Fu, 2018*; *Wu et al., 2020*). *Li et al. (2022)* showed excellent reliability in quantifying lumbar erector spinae stiffness in patients with CLBP in prone and sitting positions. *Feng, Li & Zhang (2019)* also investigated changes in muscle stiffness in healthy volunteers in the static prone position, sitting and upright standing. These results suggest that it is helpful to investigate the modulation of the stiffness of the erector spinae muscle of the lumbar spine in different postures to prevent CLBP.

There are currently few studies evaluating the reliability of an instrument for measuring lumbar muscle stiffness in prone or sitting position. In line with the results of previous studies in this field (*Feng, Li & Zhang, 2019*; *Lohr et al., 2018*), most MyotonPRO measurements were performed in the prone position, both in healthy individuals and in individuals suffering from low back pain. The aim our study was to evaluate the intra-rater and inter-reliability of lumbar erector spinae muscle stiffness with the MyotonPRO in healthy adults in prone and sitting position and to compare the changes in lumbar spine stiffness between prone and sitting position for the right and left side. Our hypothesis was that the intra-rater and inter-reliability of lumbar erector muscle stiffness measured with the MyotonPRO in healthy adults is good to excellent in both prone and sitting positions. Using the sitting position to assess lumbar tissue tension could be a valuable alternative to the prone position, especially for patients who are uncomfortable in the prone position and it could have further practical implications for clinical practice.

# MATERIALS AND METHODS

## Study design

This study aimed to evaluate the inter-rater and intra-rater reliability of the assessment of myofascial stiffness of the erector spinae muscle in prone and sitting position using a MyotonPro handheld myotonometer in healthy subjects.

**Table 1 Descriptive statistics of the participants.**

|  | Age (years) | Body height (cm) | Body mass (kg) | BMI (kg/m$^2$) |
|---|---|---|---|---|
| Mean | 39.8 | 174.5 | 72.3 | 23.5 |
| Median | 40 | 175 | 76.5 | 23.4 |
| Standard deviation | 10 | 10.5 | 14.7 | 2.9 |
| Min–Max | 21–61 | 153–194 | 50–59 | 18.4–29.7 |

## Ethics statements

The current study was carried out by the Declaration of Helsinki and was authorized by the Republic of Slovenia National Medical Ethics Committee (No. 0120-520/2022/3). The research was conducted at the University Medical Centre of Ljubljana's Institute for Medical Rehabilitation and Clinical Department of Neurosurgery. Prior to participation, all volunteers have read an information about the purpose of the study and have provided written informed consent.

## Sample

Thirty asymptomatic participants ($N$ = 15 women and $N$ = 15 men) between 18 and 65 years were recruited to participate in this study. They were invited through electronic media (email, Facebook, Whatsapp, Viber, *etc.*) and chain referral/snowball sampling was used Demographics and anthropometrics of the participants are reported in Table 1.

## Inclusion exclusion criteria

Participants were recruited if they were in good health and had no musculoskeletal lower back pain in the previous week and on the day of the testing. Exclusion criteria included previous spinal surgery, spinal deformities (such as scoliosis or kyphosis), osteoporosis, lumbar disc protrusion, pregnancy and a body mass index (BMI) of 30 kg/m$^2$ or higher (*Nair et al., 2016*). Participants with significant spinal or other pathological conditions, malignant and systemic diseases or neurological, respiratory, cardiovascular and orthopaedic disorders that could have an impact on the test results were excluded from the study. In addition, alcohol consumption and excessive physical activity within the last 24 h before the test, which could lead to dehydration and increased stiffness of the fascial tissue, were considered reasons for exclusion.

## Study instruments

The stiffness of the lumbar erector spinae muscles was measured using a non-invasive, portable MyotonPro device (*Myoton Ltd, 2016*). The device captured the reduced natural vibration of the soft tissue as an acceleration signal to evaluate the biomechanical properties of stiffness, frequency, damping, deformation over time and stress relaxation time from the acceleration and displacement signals. The stiffness measurements were performed with the MyotonPro device by Examiner 1, an experienced physician who has been using the device in daily scientific research for 2 years, and Examiner 2, an experienced physiotherapist with over 20 years of practice in musculoskeletal, fascial and

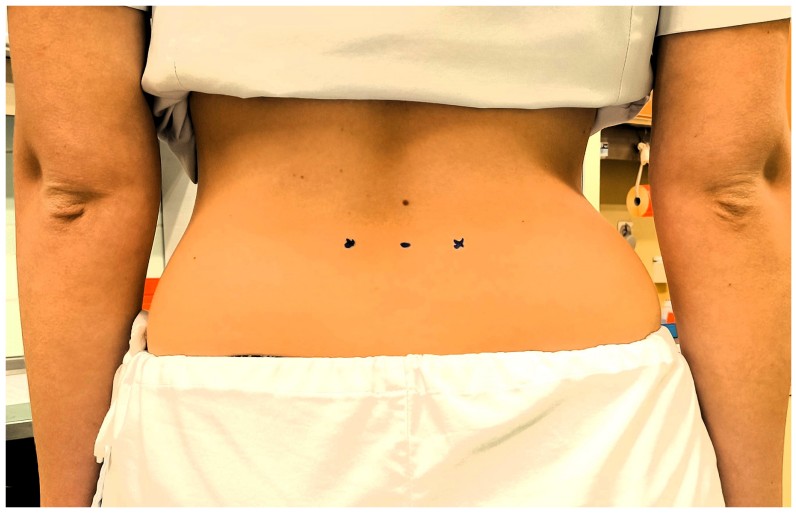

**Figure 1** **The measurement site of the erector spinae muscles.**

neurological rehabilitation. Before starting the examination, both examiners practised using the device on the lumbar erector spinae. Prior to the actual data collection, a pilot test was conducted with a sample of ten subjects using the MyotonPro device. This was done to improve the testing skills of the examiners and to refine the protocol. Both examiners are right-handed. The inter-rater reliability was considered as the assessment results from two examiners on the same day, and the intra-rater reliability was calculated by the assessment results from the same examiner with an exact 24-h interval for all participants.

Measurements were first taken in a resting prone position and then in a sitting position on both sides (first on the left side and then on the right muscle belly at the level of L4 vertebra). The palpable belly of the muscle one finger's width from the spinous process at the level of L4 was determined as the location for the measurements (*Lohr et al., 2018*). The exact position was confirmed and marked with a permanent marker (Fig. 1). All measurements were performed in a designated room where the temperature was kept at a stable level of approximately 25 °C for all participants.

## Measurement of the erector spinae stiffness in different positions

When the subjects arrived, they were asked to rest and relax for 10 min at room temperature to normalise body conditions. Participant demographics were recorded before the experiment began. The protocol was shown to all participants in turn. First, the experienced physician (Examiner 1) performed the measurements, then the experienced physiotherapist (Examiner 2) performed the tests. Each subject was asked to lie prone on an examination table and relax for 5 min with their arms resting at their sides before the procedure began. To ensure greater relaxation, a foam pad was placed under the ankles to ensure a neutral, relaxed foot position. During the procedure, all participants were asked to hold their breath for 5 s at the end of inhalation to minimise the confounding factor resulting from the intra-abdominal pressure changes that occur during natural breathing cycles. Muscle stiffness was measured on the palpable muscle belly, one finger width from

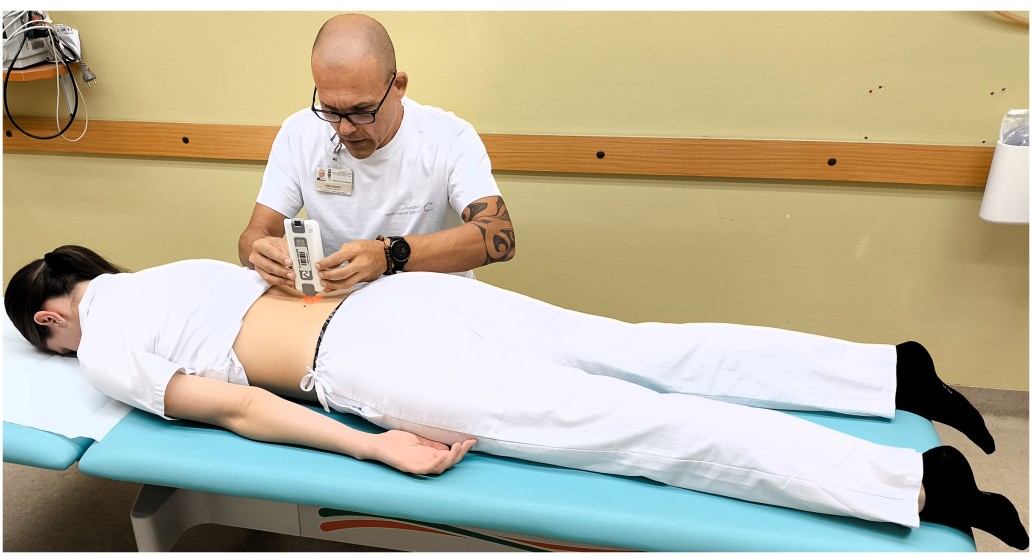

**Figure 2 Measuring the erector spinae muscle stiffness in prone position.**

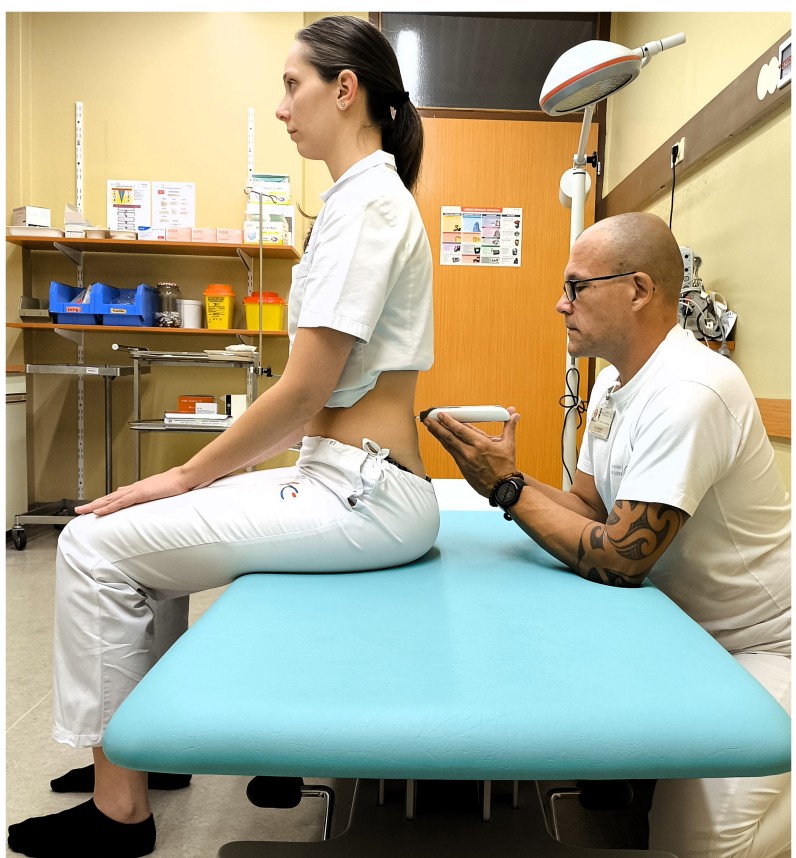

**Figure 3 Measuring the erector spinae muscle stiffness in sitting position.**

the spinous process at the level of L4 vertebra on both sides in both prone (Fig. 2) and sitting (Fig. 3) positions. The abdominal muscles of the left erector spinae were measured from the left side of the participant, the abdominal muscles of the right erector spinae from the right side. Due to the relatively large sample size and the fact that both sides were measured, it was impossible for the investigators to memorise the results. The measurements were recorded on a separate device so that the examiners were blind to each other's measurements. For the second measurement, the subject was asked to sit in a neutral position on an examination table, with the head in a neutral position and the feet on the floor. To ensure inter-rater reliability, (Examiner 1 and Examiner 2) performed repeated measurements in prone and sitting positions 30 min apart. In addition, Examiner 2 repeated the test protocol after a 24-h interval to confirm intra-rater reliability. The total duration of the test was 1 h for each measurement, with half an hour for the interrater test and half an hour for each intrarater test.

## Statistical procedures

The data analysis was carried out using an Excel programme (Microsoft Corporation, Redmond, WA, USA) and IBM SPSS Statistics 22 (IBM, Armonk, NY, USA). Descriptive statistics were used to summarise the demographic data set, presenting key trends and variability through means and standard deviations. Intraclass correlation coefficients (ICCs) together with 95% confidence intervals (CI) were used to assess reliability. This allowed the assessment of both within-session reliability using a one-sided random model and between-subject reliability using a two-sided random model. The test reliability level measured by the ICC was determined according to the classification of *Portney & Watkins (2009)* and *Domholdt (1993)* using the following criteria: excellent (0.90–1.00), good (0.70–0.89), moderate (0.50–0.69) and poor (<0.49). The standard error of measurement (SEM) was calculated using the formula SEM = standard deviation × $\sqrt{1-ICC}$. The minimum detectable change (MDC) was calculated using the formula MDC = 1.96 × SEM × $\sqrt{2}$. SEM% was defined as SEM% = (SEM/mean) × 100 and MDC% as MDC% = (MDC/mean) × 100. Visual representations were created using Bland-Altman plots to illustrate the level of agreement. Independent t-tests were performed separately to compare the stiffness of the erector spinae on the left and right sides in different positions, especially in prone and sitting position.

## RESULTS

Intra-rater reliability was assessed using a one-way random model consistency. Measurements of left erector spinae muscle stiffness in the prone position showed intra-rater reliability with ICC of 0.88 (95% CI [0.76–0.94]), SEM of 28.67 N/m and MDC of 79.47 N/m. Similarly, the right side in prone position showed intra-rater reliability with ICC of 0.91 (95% CI [0.82–0.95]), SEM of 24.13 N/m and MDC of 66.88 N/m. The intra-rater reliability percentages for SEM and MDC, for testing in the prone position were both less than 8.87% and 24.61% for the left and right sides, respectively.

The intra-rater reliability for the left erector spinae muscle testing in sitting position indicated ICC of 0.91 (95% CI [0.82–0.96]), SEM of 43.30 N/m and MDC of 120.03 N/m.

**Table 2 Erector spinae muscle stiffness measurements.**

**Intra-rater reliability**

| N = 30 | TEST 1 (mean ± SD) (N/m) | TEST 2 (mean ± SD) (N/m) | ICC | 95% IC | SEM | SEM (%) | MDC | MDC (%) |
|---|---|---|---|---|---|---|---|---|
| Prone left | 317.00 +/− 70.18 | 328.97 +/− 106.03 | 0.88 | [0.76–0.94] | 28.67 | 8.87 | 79.47 | 24.61 |
| Prone right | 325.33 +/− 82.50 | 341.00 +/− 88.41 | 0.91 | [0.82–0.95] | 24.13 | 7.24 | 66.88 | 20.06 |
| Sitting left | 493.33 +/− 173.56 | 479.97 +/− 139.32 | 0.91 | [0.82–0.96] | 43.30 | 8.90 | 120.03 | 24.66 |
| Sitting right | 554.60 +/− 173.45 | 497.87 +/− 142.07 | 0.89 | [0.78–0.95] | 49.16 | 9.35 | 136.27 | 25.91 |

**Inter-rater reliability**

| N = 30 | Examiner 1 (mean ± SD) (N/m) | Examiner 2 (mean ± SD) (N/m) | ICC | 95% IC | SEM | SEM (%) | MDC | MDC (%) |
|---|---|---|---|---|---|---|---|---|
| Prone left | 315.50 +/− 84.69 | 317.00 +/− 70.18 | 0.87 | [0.73–0.94] | 26.11 | 8.26 | 72.37 | 22.89 |
| Prone right | 340.63 +/−115.81 | 325.33 +/− 82.50 | 0.84 | [0.68–0.92] | 36.43 | 10.94 | 100.96 | 30.32 |
| Sitting left | 497.17 +/− 175.74 | 493.33 +/− 173.56 | 0.96 | [0.92–0.98] | 32.55 | 6.58 | 90.24 | 18.24 |
| Sitting right | 544.13 +/− 167.60 | 554.60 +/− 173.45 | 0.95 | [0.90–0.97] | 35.39 | 6.44 | 98.11 | 17.85 |

On the right side, the intra-rater reliability indicated ICC of 0.89 (95% CI [0.78–0.95]), SEM of 49.16 N/m and MDC of 136.27 N/m. The intra-rater reliability percentages for SEM and MDC, for testing in the sitting position, were both less than 9.35% and 25.91% for the right and left sides, respectively.

Inter-rater reliability was analyzed using a two-way random model consistency. In the prone position, inter-rater reliability demonstrated ICC of 0.87 (95% CI [0.73–0.94]), SEM of 26.11 N/m and MDC of 72.37 N/m, on the left side. Similarly, on the right-side, the inter-rater reliability demonstrated ICC of 0.84 (95% CI [0.68–0.92]), SEM of 36.43 N/m, and MDC of 100.96 N/m. The inter-rater reliability percentages for SEM and MDC, for testing in the prone position, were both less than 10.94% and 30.32% for the right and left sides, respectively.

In terms of inter-rater reliability for the left erector spinae muscle testing in sitting position, the ICC with 95% CI was 0.96 (0.92–0.98), SEM of 32.55 N/m, and MDC of 90.24 N/m. Similarly, for the right sitting position, the ICC with 95% CI was 0.95 (0.90–0.97), with SEM of 35.39 N/m and MDC of 98.11 N/m. The inter-rater reliability percentages for SEM and MDC for testing in the sitting position were both less than 6.58% and 18.24% for the right and left sides, respectively.

Table 2 summarises the intra-rater and inter-rater reliability of erector spinae stiffness measured with MyotonPRO. Tests 1 and 2 represent the results of examiner 1, which measured the erector spinae muscles first on the left and then on the right side

in prone and then sitting position. The time interval between the first and the following measurement was 24 h. Similarly, the inter-rater reliability of the MyotonPRO was demonstrated by summarising the results of a test performed on the same day with a 30-min interval between the first and repeated test of examiner 1 and 2 in each person tested.

Additionally, Figs. 4–7 display the Bland-Altman plots to visually represent the degree of agreement. Figure 8 represents bias of measurement between different examiners.

The data in Tables 3 and 4 show the mean muscle stiffness, the paired difference analysis, and the effect sizes between prone and sitting positions for the right and left erector spinae muscle stiffness in healthy populations.

The data in Table 3 show significant differences in tissue tension between the prone and sitting positions, with the sitting position showing a higher level of muscle stiffness. The mean muscle stiffness in the prone position is 328.07 N/m, with a standard deviation of 83.53 N/m. On the other hand, the mean stiffness of the erector spinae muscle in the sitting position is significantly higher (511.18 N/m), with a standard deviation of 511.23 N/m. The paired differences analysis shows that the mean difference in tissue tension between the prone and sitting positions is −183.11 N/m with a standard deviation of 99.03 N/m. This difference is statistically significant, with a t-value of −10.13 and a two-tailed $p$-value of less than 0.001. In addition, effect size analysis using Cohen's d and Hedges' correction indicates a significant difference in tissue stiffness between the two positions. Cohen's d is calculated as −1.84, while Hedges' correction gives a value of −1.80. Both effect size measures indicate a large effect, highlighting the significant difference in tissue stiffness between the prone position and sitting position.

Analysis of the paired samples revealed significant differences in the stiffness of the erector spinae muscle between the left and right sides. The results were determined using the mean values of all measurements taken in both positions and by both examiners. On the left side, the mean erector spinae muscle stiffness was 405.32 N/m with a standard deviation of 112.65 N/m, while on the right side, it was slightly higher at 433.92 N/m with a standard deviation of 115.47 N/m. A strong positive correlation of 0.91 was observed between the tissue tension values on both sides, which was statistically significant ($p < 0.001$). Analysis using the paired samples test showed a mean difference in tissue tension between the left and right sides of −28.60 N/m with a standard deviation of 46.51 N/m. This difference was found to be statistically significant, supported by a t-value of −3.36 and a two-sided $p$-value of 0.002. Effect size analysis using Cohen's d and Hedges' correction indicated a moderate effect size. Cohen's d was calculated to be −0.61, while Hedges' correction yielded a value of −0.59. These effect sizes suggest a moderate difference in erector muscle stiffness between the left and right sides. In summary, these results show statistically significant differences in tissue tension between the left and right sides, with the right side showing slightly higher tension levels on average.

## DISCUSSION

The primary objective of this study was to determine the intra-rater and inter-reliability of measuring myofascial stiffness of erector spinae muscle, using MyotonPRO in healthy

**Prone left: Inter-rater reliability**

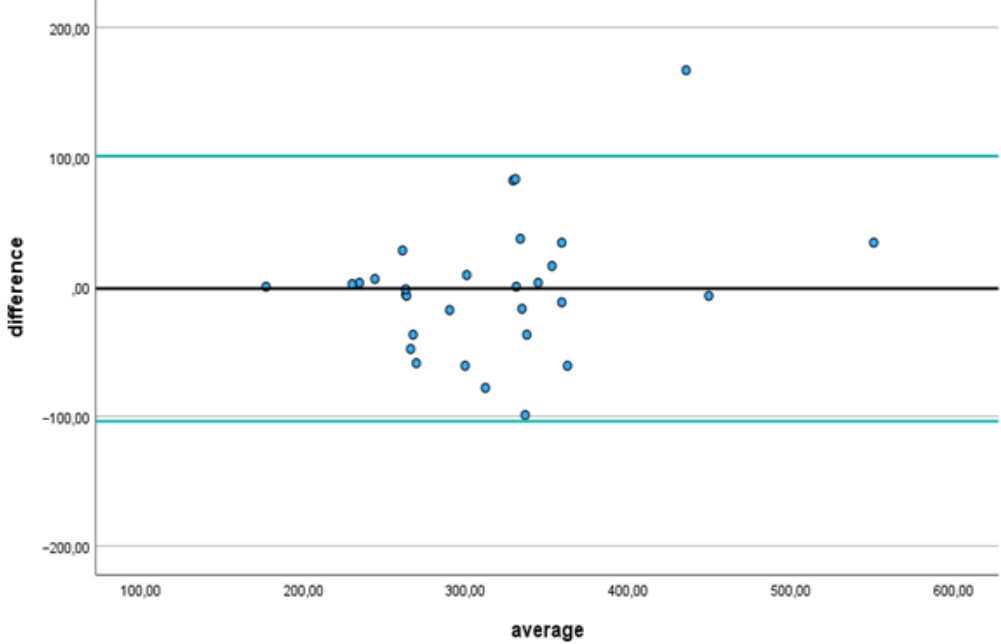

**Prone left: Intra-rater reliability**

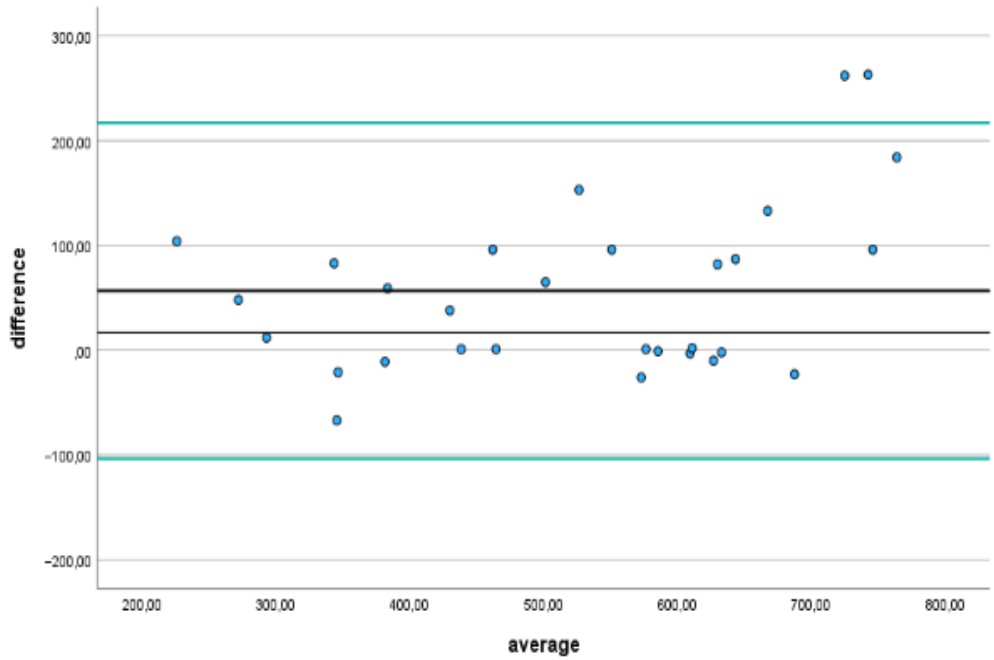

**Figure 4  Bland Altman plots.** Limits of agreement for left prone positions are represented by a black line and green solid lines, respectively.   

adults in both prone and sitting position. We were also able to compare myofascial stiffness of erector spinae muscle between the right and left sides and the changes in stiffness in the prone and sitting positions in a healthy population. The results of the study

**Prone right: Inter-rater reliability**

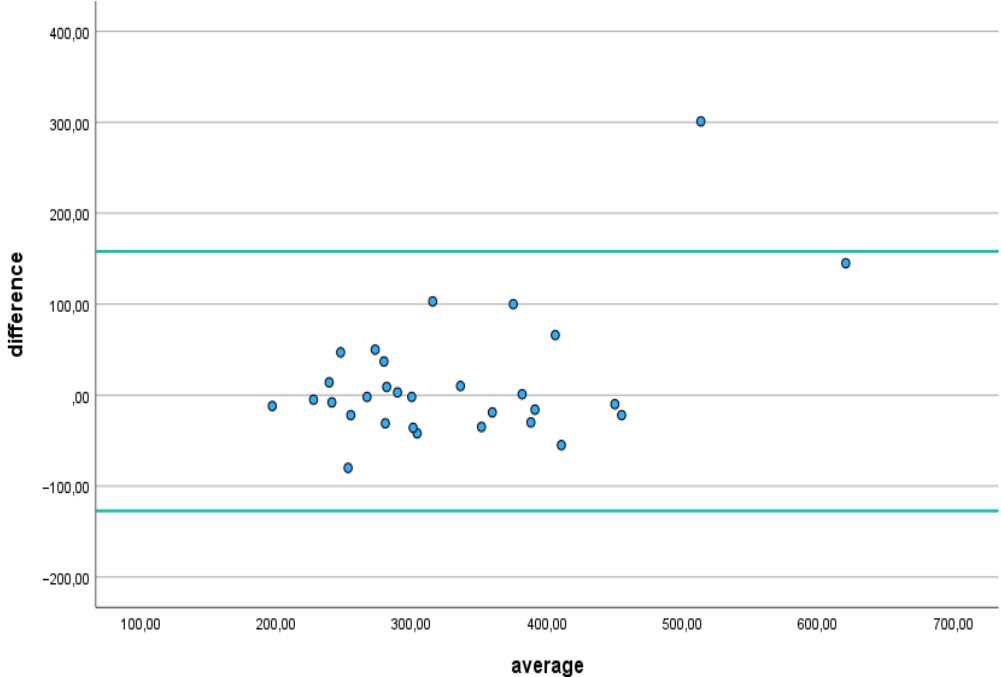

**Prone right: Intra-rater reliability**

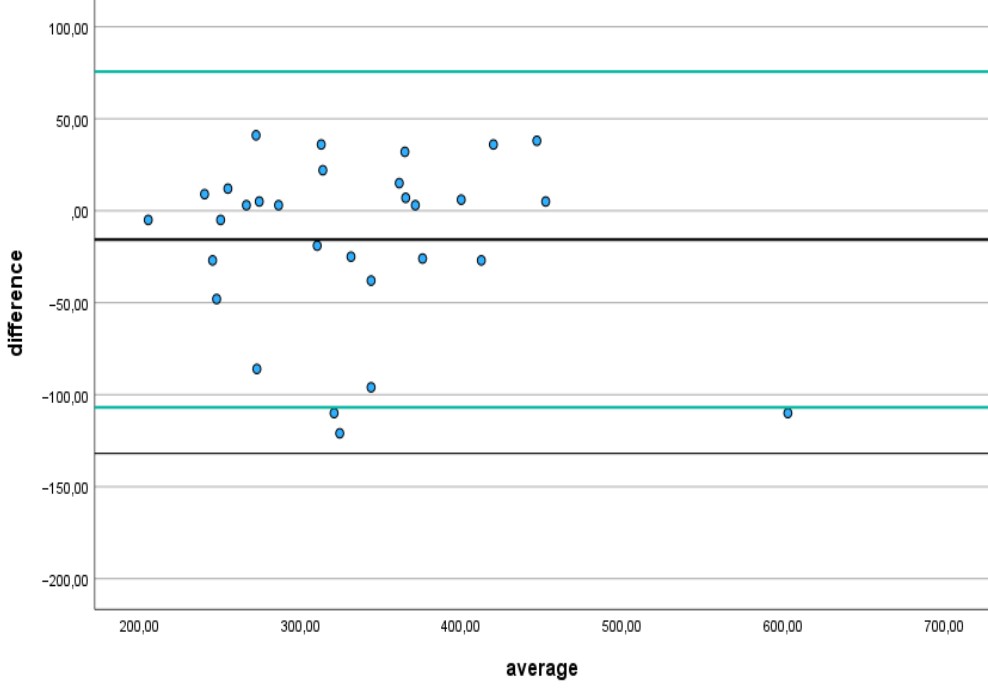

**Figure 5** **Bland Altman plots.** Limits of agreement for right prone positions are represented by a black line and green solid lines, respectively.
**Sitting left: Inter-rater reliability**

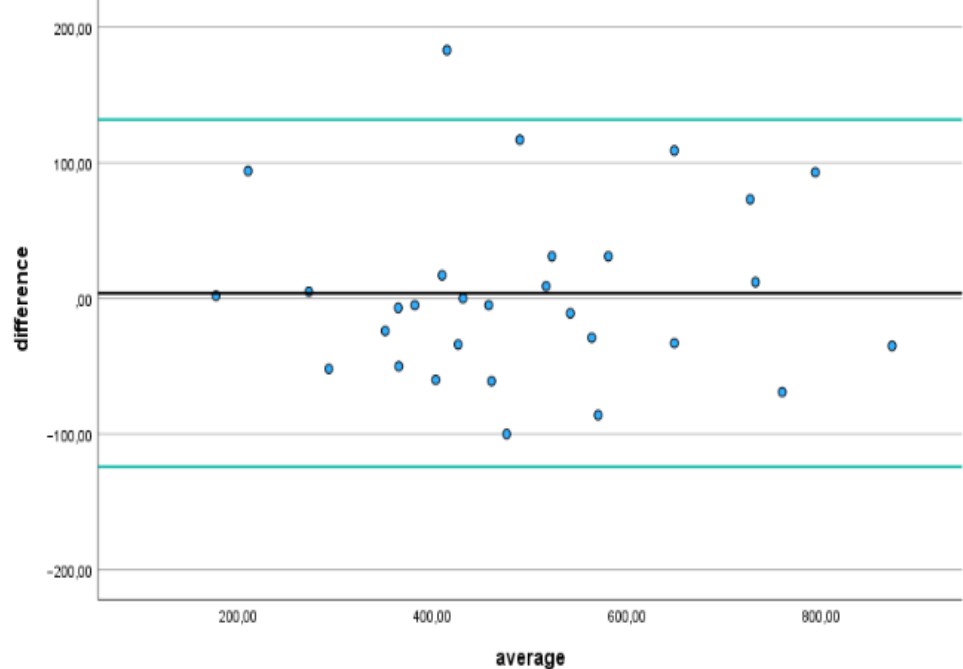

**Sitting left: Intra-rater reliability**

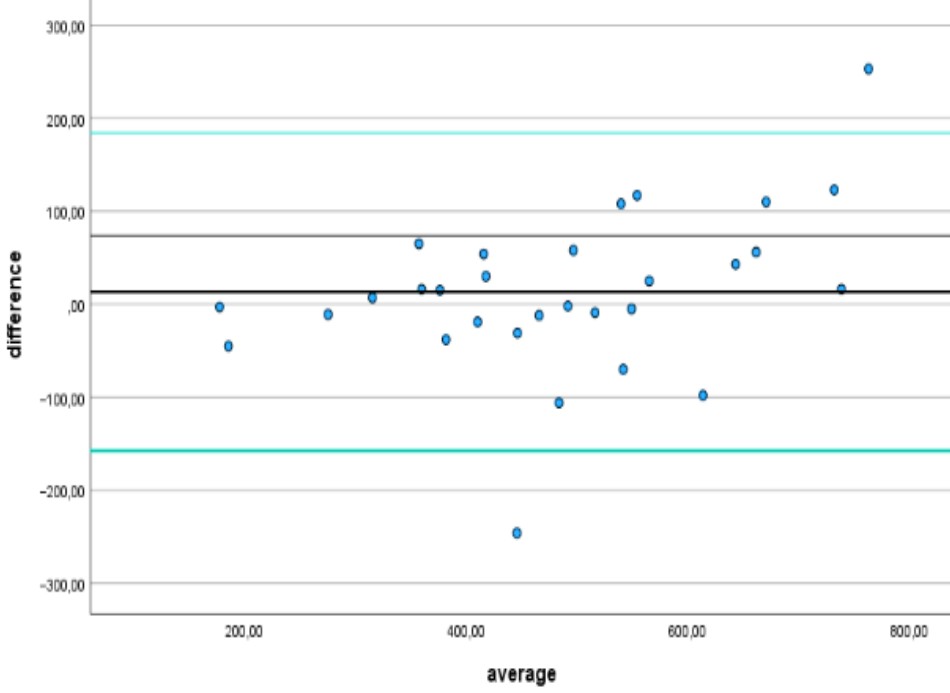

**Figure 6 Bland Altman plots.** Limits of agreement for left sitting positions are represented by a black line and green solid lines, respectively.

**Sitting right: Inter-rater reliability**

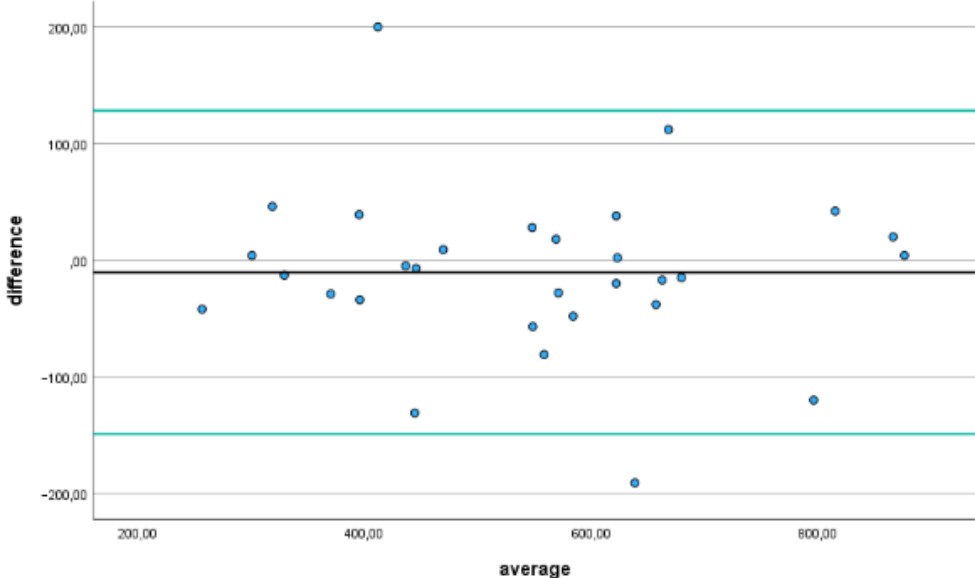

**Sitting right: Intra-rater reliability**

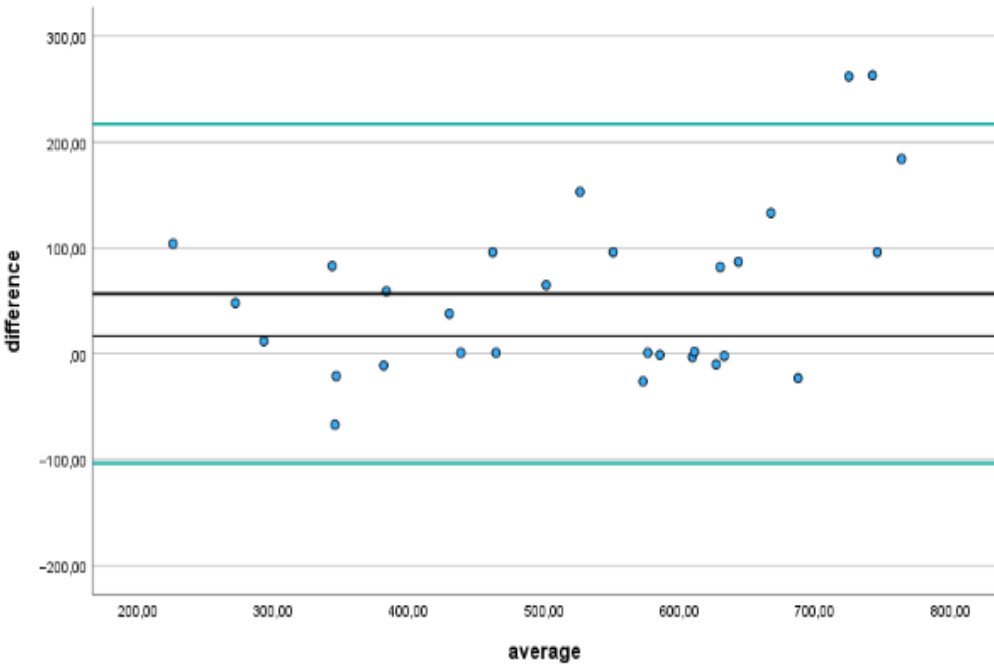

**Figure 7 Bland Altman plots.** Limits of agreement for right sitting positions are represented by a black line and green solid lines, respectively.

indicated good to excellent intra-rater reliability in measuring myfascial stiffness for both the left (ICC = 0.76–0.94) and right (ICC = 0.82–0.95) erector spinae muscle in the prone position. For sitting position, excellent intra-rater reliability was observed for the left side
**Inter-rater reliability for the erector spinae stiffness**

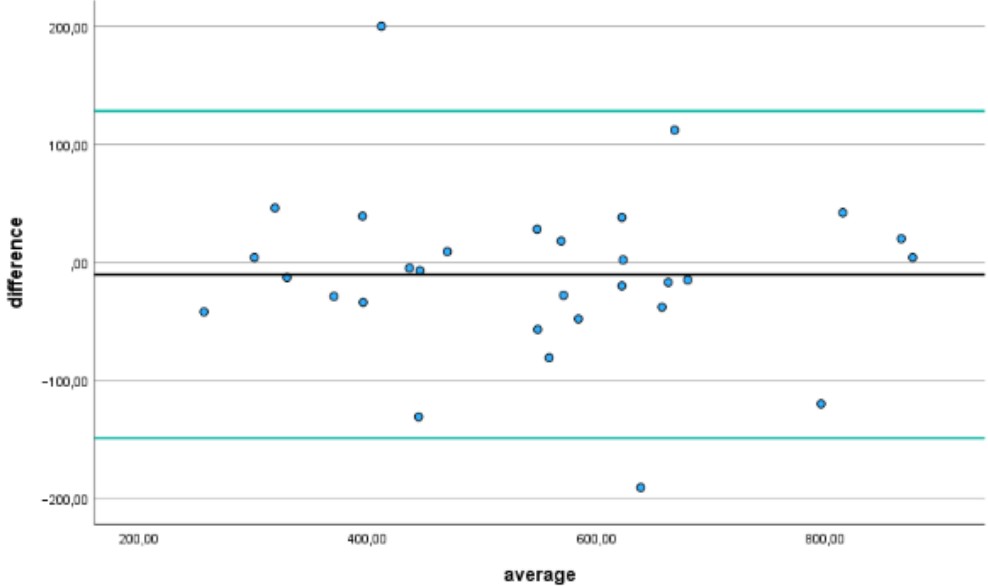

**Intra-rater reliability for the erector spinae stiffnes**

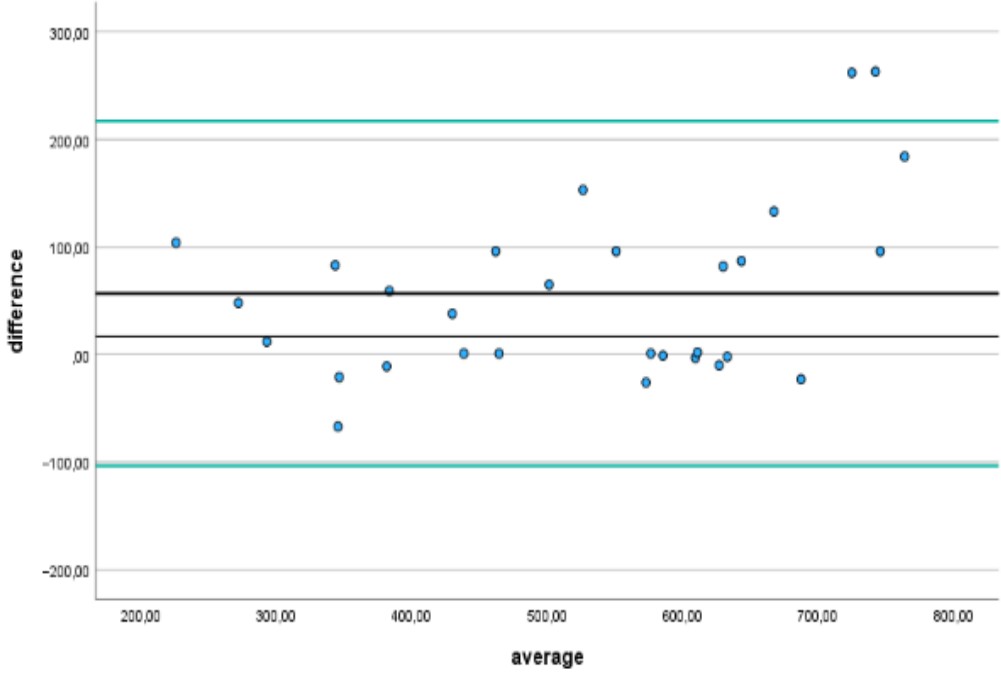

**Figure 8 Bias of measurement between different examiners.** Limits of agreement are represented by a black line and green solid lines respectively.

(ICC = 0.82–0.96), and good to excellent reliability was observed for the right side (ICC = 0.78–0.95). Inter-rater reliability showed strong agreement between raters for both prone and sitting position. The ICC values demonstrate a high degree of consistency between measurements taken by different raters. Within the range of highly reliable values,

**Table 3 Mean muscle stiffness, paired differences analysis, and effect sizes between prone and sitting position in healthy populations.**

| Position | Mean tissue stiffness (± SD) (N/m) | Sample size (N) | Mean tissue stiffness left (N/m) | Mean tissue stiffness right (N/m) |
|---|---|---|---|---|
| Prone | 328.07 (±83.53) | 30 | 320.48 | 335.65 |
| Sitting | 511.18 (±151.23) | 30 | 490.15 | 532.20 |

| Paired differences | Mean (N/m) | Std. deviation (N/m) | Std. error mean (N/m) | 95% CI lower | 95% CI upper | t-value | df | One-sided p | Two-sided p |
|---|---|---|---|---|---|---|---|---|---|
| Prone/Sitting | −183.11 | 99.03 | 18.08 | −220.08 | −146.13 | −10.13 | 29 | <0.001 | <0.001 |

| Effect sizes | Standardizer[a] | Point estimate | 95% CI lower | 95% CI upper |
|---|---|---|---|---|
| Prone/Sitting Cohen's d | 99.03 | −1.84 | −2.43 | −1.24 |
| Prone/Sitting Hedges' correction | 101.68 | −1.80 | −2.37 | −1.21 |

**Note:**
[a] The denominator used in estimating the effect sizes. Cohen's d uses the sample standard deviation of the mean difference. Hedges' correction uses the sample standard deviation of the mean difference, plus a correction factor.

**Table 4 Summary of the mean erector muscle stiffness, paired differences analysis, and effect sizes between right and left side in healthy population.**

| Position | Mean tissue stiffness (± SD) | Sample size (N) | Mean tissue stiffness prone | Mean tissue stiffness sitting |
|---|---|---|---|---|
| Left side | 405.32 (±112.65) | 30 | 320.48 | 490.15 |
| Right side | 433.92 (±115.47) | 30 | 335.65 | 532.20 |

| Paired differences | Mean | Std. deviation | Std. error mean | 95% CI lower | 95% CI upper | t-value | df | One-sided p | Two-sided p |
|---|---|---|---|---|---|---|---|---|---|
| Left/Right | −28.60 | 46.51409 | 8.49227 | −45.97 | −11.23 | −3.36 | 29 | <0.001 | <0.001 |

| Effect sizes | Standardizer[a] | Point estimate | 95% CI lower | 95% CI upper |
|---|---|---|---|---|
| Left/Right Cohen's d | 46.51 | −0.61 | −1.00 | −0.22 |
| Left/Right Hedges' correction | 47.76 | −0.59 | −0.97 | −0.21 |

**Note:**
[a] The denominator used in estimating the effect sizes. Cohen's d uses the sample standard deviation of the mean difference. Hedges' correction uses the sample standard deviation of the mean difference, plus a correction factor.

some values may deviate from the average due to variations in myofascial stiffness between measurements or from 1 day to the next due to muscle activity. Our reliability results are comparable to those of previous studies using MyotonPRO in healthy populations, showing high ICC values for test-retest intervals testing in prone position ranging from (ICC = 0.75 to 0.99), indicating good to excellent reliability (*Lohr et al., 2018*). In addition, analysis of the biomechanical properties of the lumbar extensor myofascia in healthy
individuals and elderly patients with CLBP also showed high reliability between different examiners, with measurements of muscle tone, stiffness, and elasticity of the left and right extensor myofascial tissues also tested in prone position ranging from ICC = 0.90 to ICC = 0.95 (*Wu et al., 2020*). Currently, only a few studies have evaluated the reliability of MyotonPRO for measuring lumbar myofascial stiffness in a pathological state, either in prone and sitting position. However, there are no studies evaluating reliability in healthy populations, only in sitting position. These methodological differences underscore the importance of careful interpretation when synthesizing findings across studies. Such variations in measurement techniques may influence the overall understanding of back myofascial stiffness in individuals with and without low back pain as pointed out in the systematic review from *Vatovec & Voglar (2024)*. Their analysis of pooled data highlighted notable differences in research methodologies. For instance, *Wu et al. (2022)* examined muscle tone (measured in Hz) and stiffness (measured in N/m) in paravertebral muscles at each level from L1 to L5, with participants positioned prone. In contrast, *Ilahi et al. (2020)* investigated five biomechanical properties of stiffness—frequency, decrement, creep, and stress relaxation time—in the L3–L4 myofascial tissue. Their study focused on individuals with chronic low back pain (CLBP) and matched normal controls, evaluating both left and right sides in a prone position. Furthermore, *Alcaraz-Clariana et al. (2021)* contributed to the diversity of methodologies by examining stiffness and tone, specifically in the erector spinae muscles at the L5 level, also with participants in a prone position. However, all these methodological differences in our results expose that myotonometry can reliably detect changes in myofascial stiffness of erector spinae muscle when measured in both prone and sitting positions in healthy populations. *Wu et al. (2022)* also demonstrated excellent intra- and inter-rater reliability (ICC = 0.88–0.99) tested in patients with (CLBP). *McGowen et al. (2024)* reported good to excellent test-retest reliability of stiffness measures in Baylor University Army Cadets, measuring the lumbar multifidi and longissimus thoracis muscles in standing (ICC = 0.81–0.98) and squatting (ICC = 0.93–0.96). *Hu et al. (2018)* explained that the reliability of the MyotonPro is also due to the lumbar level at which the measurements are taken. In the upper lumbar levels (L1–L2), measurements are less reliable than in the lower lumbar levels (L4) due to the attachment of the diaphragm at L1 and L2, which may affect the tone and stiffness of the paraspinal muscles during the respiratory cycle. This study used the SEM to estimate the distribution of repeated measures around the 'true' score, while the MDC reflects the smallest amount of true change rather than the measurement error inherent in the score (*Lin et al., 2009*). Intra-rater reliability SEM and MDC values were less than 8.87% and 24.6%, for the right and left side, respectively, in the prone position, and less than 9.35% and 25.9%, for the right and left side, respectively, in the sitting position. For the assessment of inter-rater reliability, both the standard error of measurement (SEM) and minimal detectable change (MDC) percentages were less than 10.94% and 30.32% in the prone position and less than 6.58% and 18.24% in the sitting position. The reported values indicate that the measurements were reliable with minimal inherent error. This suggests that the measurements obtained in both the prone and sitting positions were consistent and accurate. Bland-Altman analyses were conducted to identify systematic bias and compare

the 95% limits of agreement between the testing sessions when using the MyotonPRO to measure lumbar erector spinae stiffness in healthy participants in both prone and sitting position. Bland-Altman analyses have an advantage in that scatter plots can be used to visually interpret data from the observations of any outliers, bias, or relationship between variance in measures, size of the mean, and limits of agreement (*Chuang et al., 2013*; *Li et al., 2022*). In our study, the 95% CI of the mean difference included 0, which confirmed good repeatability. The results of this study showed that healthy participants had greater erector spinae muscle stiffness on the dominant right side (335.7 N/m) than on the non-dominant left side (320.5 N/m) measured in the prone position. The right side showed 4.73% more stiffness than the left side. Even in the sitting position, the right side had higher values (532.2 N/m) than the left side (490.2 N/m). The right side had a higher stiffness of 8.57%. *Hu et al. (2018)* found no significant differences in paraspinal muscle stiffness between the left and right sides in young adults with (CLBP) in the prone position (left = 280.9 N/m and right = 289.7 N/m). In a study by *Becker et al. (2018)*, it was found that patients with chronic low back pain (CLBP) exhibited significantly increased activity in the lumbar erector spinae when transitioning from sitting to standing during 30 s of standing and while climbing stairs. *Li et al. (2022)* argue that the human musculoskeletal system is in a state of balance and left-right symmetry when healthy. However, incorrect postures can cause alterations in muscle tone, resulting in asymmetry and postural problems. In their study of healthy individuals, the researchers found no difference in the rigidity of the erector spinae muscles on both sides in prone, sitting, and standing position. However, our study revealed that the rigidity of the erector spinae muscles might vary depending on right-hand dominance. These differences were observed in both prone and sitting position. Overall, the results suggest that the use of MyotonPRO maintains high levels of reliability in different positions, highlighting its practical and portable utility for assessing muscle stiffness in clinical practice and is better than assessments based only on palpation or observation of posture by the clinicians. This study has certain limitations. Firstly, the lumbar erector spinae muscles consist of several small muscles that lie between fascial planes. Therefore, the measurement of muscle stiffness taken at the palpable muscle belly, one finger width from the spinous process at the level of L4, may not reflect the actual stiffness of all adjacent and deeper structures. Secondly, muscle stiffness in older people may differ from that in young people, as analysed in previous studies (*Eby et al., 2015*; *Ikezoe et al., 2012*). Third, to test possible differences between men and women in the stiffness of the erector spinae muscle, as has been done by *Taş & Salkın (2019)*, by testing the stiffness of the Achilles tendon and gastrocnemius muscle at rest and under tension. Fourth, our testing protocol was designed to be easily replicated in clinical settings for patients with LBP who have difficulty lying in the prone position and are more comfortable in the sitting position. Therefore, further research should be conducted to determine whether differences in testing protocol, such as measuring the right side first rather than the left side, have an impact on test results. However, in our study we only analysed muscle stiffness. Therefore, future studies should analyse all parameters determined by the Myoton PRO, including skin oscillation frequency, logarithmic decline, relaxation time and creep, to ensure even greater reliability of the device.

## CONCLUSION

This study demonstrated high intra- and inter-reliability of lumbar erector muscle stiffness with the MyotonPRO in healthy adults and showed the device's ability to detect even small changes in M erector spine stiffness, testing both right and left sides and measuring in both prone and sitting position. Using the sitting position to assess lumbar stiffness could be a useful alternative to the prone position, particularly for patients who are uncomfortable in the prone position. This could have further practical implications for clinical setting.

## ACKNOWLEDGEMENTS

We thank all the participants for their cooperation.

### Funding

The Slovenian Research Agency supported this work (research core funding No. P3-0388). The funders had no role in study design, data collection and analysis, decision to publish, or preparation of the manuscript.

### Grant Disclosures

The following grant information was disclosed by the authors:
The Slovenian Research Agency: P3-0388.

### Competing Interests

The authors declare that they have no competing interests.

### Author Contributions

- Fabio Valenti conceived and designed the experiments, performed the experiments, analyzed the data, prepared figures and/or tables, authored or reviewed drafts of the article, and approved the final draft.
- Sara Meden conceived and designed the experiments, performed the experiments, analyzed the data, prepared figures and/or tables, authored or reviewed drafts of the article, and approved the final draft.
- Maja Frangež conceived and designed the experiments, authored or reviewed drafts of the article, and approved the final draft.
- Renata Vauhnik conceived and designed the experiments, authored or reviewed drafts of the article, and approved the final draft.

### Human Ethics

The following information was supplied relating to ethical approvals (*i.e.*, approving body and any reference numbers):

The current study was carried out in accordance with the Declaration of Helsinki and was authorized by the Republic of Slovenia National Medical Ethics Committee (No. 0120-520/2022/3).

## Data Availability

The raw data are available in the Supplemental File.

## Supplemental Information

Supplemental information for this article can be found online at http://dx.doi.org/10.7717/peerj.18524#supplemental-information.

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
