# Peer review of "Intra-rater and inter-rater reliability of a handheld myotonometer measuring myofascial stiffness of lower lumbar myofascial tissue in healthy adults"

_PeerJ, doi:10.7717/peerj.18524_

## Round 0.1 · original submission · Major Revisions

Please respond to the reviews. We look forward to your revision

Reviewer 1 ·

Basic reporting

The study is interesting and the manuscript is well written. I have some minor comments:
- Some typos are present throughout the manuscript
- In the materials and methods - sample section, the authors report some data with commas to separate the decimals, instead of using the dot. I think this is a typo. Also, I think that simply reporting the mean+-standard deviation, instead of writing for each measure mean xxx SD yyyy is more readable.
- In general, there is no need to duplicate what is in the text also in the tables, and vice-versa, so I suggest removing Table 1, or removing the sentence in the methods and simply reporting that demographics and anthropometrics of the participants are reported in Table 1.
- As above, please check the use of the correct symbol for decimals, as commas are used also in the tables.
- I think that in the results section authors should limit the "interpretation" of the findings, such as the use of terms such as "notable", "excellent"

Experimental design

No comment

Validity of the findings

I think that the authors could provide some data, if present, to strengthen the validity of this device compared to other techniques to evaluate muscle stiffness, such as MRI or SWE. Indeed, it is important to understand its reliability, but it needs to be well-defined its "validity".

The authors discuss possible sex differences and cite some literature regarding Achilles tendon; it should be noted that there is already some literature about sex differences in lumbar muscles stiffness using different techniques as myoton, ultrasound or tensiomyography (Deodato et al., 2023; Rodrigues-de-Souza et al., 2023; Wu et al., 2021)

·

Basic reporting

- In terms of English writing, there are many ambiguities in the text and tables.
- Literature references, sufficient field background/context provided.
- Professional article structure, figures, tables and raw data shared. ( (The tables design are not suitable)
- Self-contained with relevant results to aims.

Experimental design

- Original primary research can be within the aims and scope of the journal.

- The research question is not well defined. The necessity of doing the work or the novelty of this research is not stated. The hypothesis of this project is not clear.

- Rigorous investigation performed to technical & ethical standard.

- Methods described with sufficient detail & information to replicate

Validity of the findings

- All underlying data have been provided; they are robust, statistically sound, & controlled.
- Conclusions are mostly related to research objectives.

Additional comments

Please reply to the comments in the attached file. Also make the requested corrections.

Reviewer 3 ·

Basic reporting

The same terminology should be used throughout the article when referring to the "seated" and "sitting" positions.

The hypotheses are not clearly stated in the introduction.

There are errors, especially in the reference section. (Line 54, 85, 87)

Experimental design

On what basis were 30 cases taken in the experimental design? No sample size calculation was made. The fact that the age range covers a large range of 18-65 years makes sample size calculation necessary.

It could have been determined, in parallel with the EMG measurements, that the subjects were completely relaxed during the measurements. This situation can be seen as a shortcoming of the study.

Validity of the findings

The unit of body mass index should be specified in the findings section (Line 117).

When numerical expressions are specified in the tables, periods should be used instead of commas (Tables 3 and 4).

Additional comments

The first paragraph of the discussion section mentions findings in the supine position, but the article is structured on intra-rater and inter-rater myotonometeric measurements in the prone and sitting positions (Line 258, 261).

Limitations of the study were not stated.

---

## Round 0.2 · accepted · Accept

All reviewer concerns were addressed. I confirm that the manuscript is ready for production.

Reviewer 1 ·

Basic reporting

I have no further comments, thank you.

Experimental design

I have no further comments, thank you.

Validity of the findings

I have no further comments, thank you.

Additional comments

I have no further comments, thank you.

·

Basic reporting

The study is interesting and the manuscript has been well rewritten. The expectations of the reviewer have been met.

Experimental design

The method of obtaining the sample size should be written in the text.

Validity of the findings

No comment

Additional comments

No comment

Reviewer 3 ·

Basic reporting

No comment

Experimental design

No comment

Validity of the findings

No comment

Additional comments

No comment